# One-Shot Imitation Learning

**Yan Duan**[†§]**, Marcin Andrychowicz**[‡]**, Bradly Stadie**[†‡]**, Jonathan Ho**[†§]**,
Jonas Schneider**[‡]**, Ilya Sutskever**[‡]**, Pieter Abbeel**[†§]**, Wojciech Zaremba**[‡]

[†]Berkeley AI Research Lab, [‡]OpenAI
[§]Work done while at OpenAI
{rockyduan, jonathanho, pabbeel}@eecs.berkeley.edu
{marcin, bstadie, jonas, ilyasu, woj}@openai.com

## Abstract

Imitation learning has been commonly applied to solve different tasks in isolation. This usually requires either careful feature engineering, or a significant number of samples. This is far from what we desire: ideally, robots should be able to learn from very few demonstrations of any given task, and instantly generalize to new situations of the same task, without requiring task-specific engineering. In this paper, we propose a meta-learning framework for achieving such capability, which we call *one-shot imitation learning*.

Specifically, we consider the setting where there is a very large (maybe infinite) set of tasks, and each task has many instantiations. For example, a task could be to stack all blocks on a table into a single tower, another task could be to place all blocks on a table into two-block towers, etc. In each case, different instances of the task would consist of different sets of blocks with different initial states. At training time, our algorithm is presented with pairs of demonstrations for a subset of all tasks. A neural net is trained such that when it takes as input the first demonstration demonstration and a state sampled from the second demonstration, it should predict the action corresponding to the sampled state. At test time, a full demonstration of a single instance of a new task is presented, and the neural net is expected to perform well on new instances of this new task. Our experiments show that the use of soft attention allows the model to generalize to conditions and tasks unseen in the training data. We anticipate that by training this model on a much greater variety of tasks and settings, we will obtain a general system that can turn any demonstrations into robust policies that can accomplish an overwhelming variety of tasks.

## 1   Introduction

We are interested in robotic systems that are able to perform a variety of complex useful tasks, e.g. tidying up a home or preparing a meal. The robot should be able to learn new tasks without long system interaction time. To accomplish this, we must solve two broad problems. The first problem is that of dexterity: robots should learn how to approach, grasp and pick up complex objects, and how to place or arrange them into a desired configuration. The second problem is that of communication: how to communicate the *intent* of the task at hand, so that the robot can replicate it in a broader set of initial conditions.

Demonstrations are an extremely convenient form of information we can use to teach robots to overcome these two challenges. Using demonstrations, we can unambiguously communicate essentially any manipulation task, and simultaneously provide clues about the specific motor skills required to perform the task. We can compare this with an alternative form of communication, namely natural language. Although language is highly versatile, effective, and efficient, natural language processing

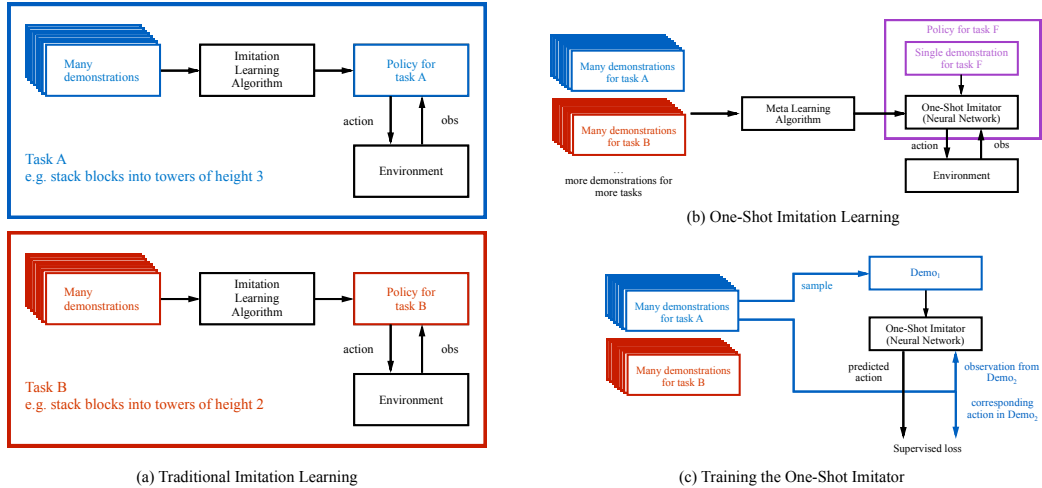

(a) Traditional Imitation Learning

(b) One-Shot Imitation Learning

(c) Training the One-Shot Imitator

Figure 1: (a) Traditionally, policies are task-specific. For example, a policy might have been trained through an imitation learning algorithm to stack blocks into towers of height 3, and then another policy would be trained to stack blocks into towers of height 2, etc. (b) In this paper, we are interested in training networks that are *not* specific to one task, but rather can be told (through a single demonstration) what the current new task is, and be successful at this new task. For example, when it is conditioned on a single demonstration for task F, it should behave like a good policy for task F. (c) We can phrase this as a supervised learning problem, where we train this network on a set of training tasks, and with enough examples it should generalize to unseen, but related tasks. To train this network, in each iteration we sample a demonstration from one of the training tasks, and feed it to the network. Then, we sample another pair of observation and action from a second demonstration of the same task. When conditioned on both the first demonstration and this observation, the network is trained to output the corresponding action.

systems are not yet at a level where we could easily use language to precisely describe a complex task to a robot. Compared to language, using demonstrations has two fundamental advantages: first, it does not require the knowledge of language, as it is possible to communicate complex tasks to humans that don't speak one's language. And second, there are many tasks that are extremely difficult to explain in words, even if we assume perfect linguistic abilities: for example, explaining how to swim without demonstration and experience seems to be, at the very least, an extremely challenging task.

Indeed, learning from demonstrations have had many successful applications . However, so far these applications have either required careful feature engineering, or a significant amount of system interaction time. This is far from what what we desire: ideally, we hope to demonstrate a certain task only once or a few times to the robot, and have it instantly generalize to new situations of the same task, without long system interaction time or domain knowledge about individual tasks.

In this paper we explore the one-shot imitation learning setting illustrated in Fig. 1, where the objective is to maximize the expected performance of the learned policy when faced with a new, previously unseen, task, and having received as input only one demonstration of that task. For the tasks we consider, the policy is expected to achieve good performance without any additional system interaction, once it has received the demonstration.

We train a policy on a broad distribution over tasks, where the number of tasks is potentially infinite. For each training task we assume the availability of a set of successful demonstrations. Our learned policy takes as input: (i) the current observation, and (ii) one demonstration that successfully solves a different instance of the same task (this demonstration is fixed for the duration of the episode). The policy outputs the current controls. We note that any pair of demonstrations for the same task provides a supervised training example for the neural net policy, where one demonstration is treated as the input, while the other as the output.

To make this model work, we made essential use of soft attention [6] for processing both the (potentially long) sequence of states and action that correspond to the demonstration, and for processing the components of the vector specifying the locations of the various blocks in our environment. The use of soft attention over both types of inputs made strong generalization possible. In particular, on a family of block stacking tasks, our neural network policy was able to perform well on novel block configurations which were not present in any training data. Videos of our experiments are available at `http://bit.ly/nips2017-oneshot`.

## 2   Related Work

Imitation learning considers the problem of acquiring skills from observing demonstrations. Survey articles include [48, 11, 3].

Two main lines of work within imitation learning are behavioral cloning, which performs supervised learning from observations to actions (e.g., [41, 44]); and inverse reinforcement learning [37], where a reward function [1, 66, 29, 18, 22] is estimated that explains the demonstrations as (near) optimal behavior. While this past work has led to a wide range of impressive robotics results, it considers each skill separately, and having learned to imitate one skill does not accelerate learning to imitate the next skill.

One-shot and few-shot learning has been studied for image recognition [61, 26, 47, 42], generative modeling [17, 43], and learning "fast" reinforcement learning agents with recurrent policies [16, 62]. Fast adaptation has also been achieved through fast-weights [5]. Like our algorithm, many of the aforementioned approaches are a form of meta-learning [58, 49, 36], where the algorithm itself is being learned. Meta-learning has also been studied to discover neural network weight optimization algorithms [8, 9, 23, 50, 2, 31]. This prior work on one-shot learning and meta-learning, however, is tailored to respective domains (image recognition, generative models, reinforcement learning, optimization) and not directly applicable in the imitation learning setting. Recently, [19] propose a generic framework for meta learning across several aforementioned domains. However they do not consider the imitation learning setting.

Reinforcement learning [56, 10] provides an alternative route to skill acquisition, by learning through trial and error. Reinforcement learning has had many successes, including Backgammon [57], helicopter control [39], Atari [35], Go [52], continuous control in simulation [51, 21, 32] and on real robots [40, 30]. However, reinforcement learning tends to require a large number of trials and requires specifying a reward function to define the task at hand. The former can be time-consuming and the latter can often be significantly more difficult than providing a demonstration [37].

Multi-task and transfer learning considers the problem of learning policies with applicability and re-use beyond a single task. Success stories include domain adaptation in computer vision [64, 34, 28, 4, 15, 24, 33, 59, 14] and control [60, 45, 46, 20, 54]. However, while acquiring a multitude of skills faster than what it would take to acquire each of the skills independently, these approaches do not provide the ability to readily pick up a new skill from a single demonstration.

Our approach heavily relies on an attention model over the demonstration and an attention model over the current observation. We use the soft attention model proposed in [6] for machine translations, and which has also been successful in image captioning [63]. The interaction networks proposed in [7, 12] also leverage locality of physical interaction in learning. Our model is also related to the sequence to sequence model [55, 13], as in both cases we consume a very long demonstration sequence and, effectively, emit a long sequence of actions.

## 3   One Shot Imitation Learning

### 3.1   Problem Formalization

We denote a distribution of tasks by $\mathbb{T}$, an individual task by $t \sim \mathbb{T}$, and a distribution of demonstrations for the task $t$ by $\mathbb{D}(t)$. A policy is symbolized by $\pi_\theta(a|o, d)$, where $a$ is an action, $o$ is an observation, $d$ is a demonstration, and $\theta$ are the parameters of the policy. A demonstration $d \sim \mathbb{D}(t)$ is a sequence of observations and actions : $d = [(o_1, a_1), (o_2, a_2), \dots, (o_T, a_T)]$. We assume that the distribution of tasks $\mathbb{T}$ is given, and that we can obtain successful demonstrations for each task. We assume that there is some scalar-valued evaluation function $R_t(d)$ (e.g. a binary value

indicating success) for each task, although this is not required during training. The objective is to maximize the expected performance of the policy, where the expectation is taken over tasks $t \in \mathbb{T}$, and demonstrations $d \in \mathbb{D}(t)$.

## 3.2  Block Stacking Tasks

To clarify the problem setting, we describe a concrete example of a distribution of block stacking tasks, which we will also later study in the experiments. The compositional structure shared among these tasks allows us to investigate nontrivial generalization to unseen tasks. For each task, the goal is to control a 7-DOF Fetch robotic arm to stack various numbers of cube-shaped blocks into a specific configuration specified by the user. Each configuration consists of a list of blocks arranged into towers of different heights, and can be identified by a string. For example, `ab cd ef gh` means that we want to stack 4 towers, each with two blocks, and we want block A to be on top of block B, block C on top of block D, block E on top of block F, and block G on top of block H. Each of these configurations correspond to a different task. Furthermore, in each episode the starting positions of the blocks may vary, which requires the learned policy to generalize even within the training tasks. In a typical task, an observation is a list of $(x, y, z)$ object positions relative to the gripper, and information if gripper is opened or closed. The number of objects may vary across different task instances. We define a *stage* as a single operation of stacking one block on top of another. For example, the task `ab cd ef gh` has 4 stages.

## 3.3  Algorithm

In order to train the neural network policy, we make use of imitation learning algorithms such as behavioral cloning and DAGGER [44], which only require demonstrations rather than reward functions to be specified. This has the potential to be more scalable, since it is often easier to demonstrate a task than specifying a well-shaped reward function [38].

We start by collecting a set of demonstrations for each task, where we add noise to the actions in order to have wider coverage in the trajectory space. In each training iteration, we sample a list of tasks (with replacement). For each sampled task, we sample a demonstration as well as a small batch of observation-action pairs. The policy is trained to regress against the desired actions when conditioned on the current observation and the demonstration, by minimizing an $\ell_2$ or cross-entropy loss based on whether actions are continuous or discrete. A high-level illustration of the training procedure is given in Fig. 1(c). Across all experiments, we use Adamax [25] to perform the optimization with a learning rate of 0.001.

# 4  Architecture

While, in principle, a generic neural network could learn the mapping from demonstration and current observation to appropriate action, we found it important to use an appropriate architecture. Our architecture for learning block stacking is one of the main contributions of this paper, and we believe it is representative of what architectures for one-shot imitation learning could look like in the future when considering more complex tasks.

Our proposed architecture consists of three modules: the demonstration network, the context network, and the manipulation network. An illustration of the architecture is shown in Fig. 2. We will describe the main operations performed in each module below, and a full specification is available in the Appendix.

## 4.1  Demonstration Network

The demonstration network receives a demonstration trajectory as input, and produces an embedding of the demonstration to be used by the policy. The size of this embedding grows linearly as a function of the length of the demonstration as well as the number of blocks in the environment.

**Temporal Dropout:** For block stacking, the demonstrations can span hundreds to thousands of time steps, and training with such long sequences can be demanding in both time and memory usage. Hence, we randomly discard a subset of time steps during training, an operation we call *temporal dropout*, analogous to [53, 27]. We denote $p$ as the proportion of time steps that are thrown away.

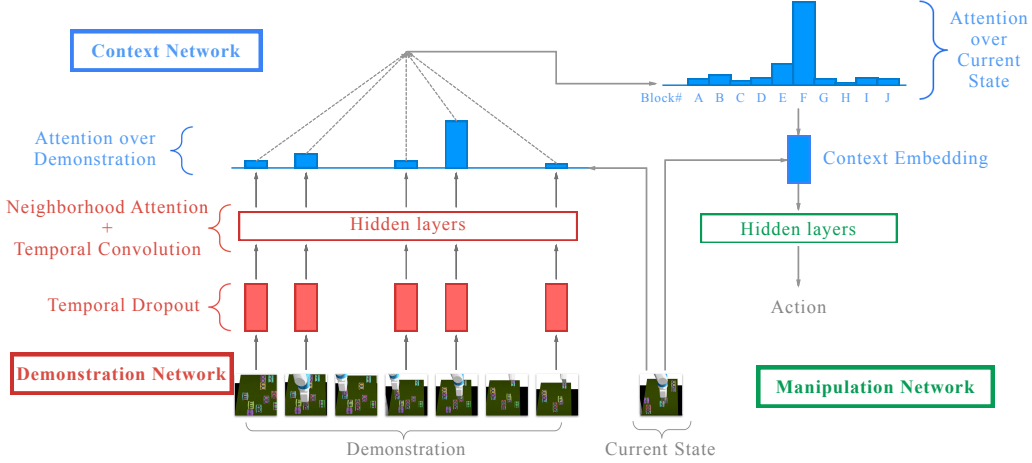

Figure 2: Illustration of the network architecture.

In our experiments, we use $p = 0.95$, which reduces the length of demonstrations by a factor of 20. During test time, we can sample multiple downsampled trajectories, use each of them to compute downstream results, and average these results to produce an ensemble estimate. In our experience, this consistently improves the performance of the policy.

**Neighborhood Attention:** After downsampling the demonstration, we apply a sequence of operations, composed of dilated temporal convolution [65] and neighborhood attention. We now describe this second operation in more detail.

Since our neural network needs to handle demonstrations with variable numbers of blocks, it must have modules that can process variable-dimensional inputs. Soft attention is a natural operation which maps variable-dimensional inputs to fixed-dimensional outputs. However, by doing so, it may lose information compared to its input. This is undesirable, since the amount of information contained in a demonstration grows as the number of blocks increases. Therefore, we need an operation that can map variable-dimensional inputs to outputs with comparable dimensions. Intuitively, rather than having a single output as a result of attending to all inputs, we have as many outputs as inputs, and have each output attending to all other inputs in relation to its own corresponding input.

We start by describing the soft attention module as specified in [6]. The input to the attention includes a query $q$, a list of context vectors $\{c_j\}$, and a list of memory vectors $\{m_j\}$. The $i$th attention weight is given by $w_i \leftarrow v^T \tanh(q + c_i)$, where $v$ is a learned weight vector. The output of attention is a weighted combination of the memory content, where the weights are given by a softmax operation over the attention weights. Formally, we have output $\leftarrow \sum_i m_i \frac{\exp(w_i)}{\sum_j \exp(w_j)}$. Note that the output has the same dimension as a memory vector. The attention operation can be generalized to multiple query heads, in which case there will be as many output vectors as there are queries.

Now we turn to neighborhood attention. We assume there are $B$ blocks in the environment. We denote the robot's state as $s_{\text{robot}}$, and the coordinates of each block as $(x_1, y_1, z_1), \ldots, (x_B, y_B, z_B)$. The input to neighborhood attention is a list of embeddings $h_1^{in}, \ldots, h_B^{in}$ of the same dimension, which can be the result of a projection operation over a list of block positions, or the output of a previous neighborhood attention operation. Given this list of embeddings, we use two separate linear layers to compute a query vector and a context embedding for each block: $q_i \leftarrow \text{Linear}(h_i^{in})$, and $c_i \leftarrow \text{Linear}(h_i^{in})$. The memory content to be extracted consists of the coordinates of each block, concatenated with the input embedding. The $i$th query result is given by the following soft attention operation: $\text{result}_i \leftarrow \text{SoftAttn}(\text{query}: q_i, \text{context}: \{c_j\}_{j=1}^B, \text{memory}: \{((x_j, y_j, z_j), h_j^{in}))\}_{j=1}^B)$.

Intuitively, this operation allows each block to query other blocks in relation to itself (e.g. find the closest block), and extract the queried information. The gathered results are then combined with each block's own information, to produce the output embedding per block. Concretely, we have

$\text{output}_i \leftarrow \text{Linear}(\text{concat}(h_i^{in}, \text{result}_i, (x_i, y_i, z_i), s_{\text{robot}}))$. In practice, we use multiple query heads per block, so that the size of each $\text{result}_i$ will be proportional to the number of query heads.

## 4.2  Context network

The context network is the crux of our model. It processes both the current state and the embedding produced by the demonstration network, and outputs a context embedding, whose dimension does not depend on the length of the demonstration, or the number of blocks in the environment. Hence, it is forced to capture only the relevant information, which will be used by the manipulation network.

**Attention over demonstration**: The context network starts by computing a query vector as a function of the current state, which is then used to attend over the different time steps in the demonstration embedding. The attention weights over different blocks within the same time step are summed together, to produce a single weight per time step. The result of this temporal attention is a vector whose size is proportional to the number of blocks in the environment. We then apply neighborhood attention to propagate the information across the embeddings of each block. This process is repeated multiple times, where the state is advanced using an LSTM cell with untied weights.

**Attention over current state**: The previous operations produce an embedding whose size is independent of the length of the demonstration, but still dependent on the number of blocks. We then apply standard soft attention over the current state to produce fixed-dimensional vectors, where the memory content only consists of positions of each block, which, together with the robot's state, forms the *context embedding*, which is then passed to the manipulation network.

Intuitively, although the number of objects in the environment may vary, at each stage of the manipulation operation, the number of relevant objects is small and usually fixed. For the block stacking environment specifically, the robot should only need to pay attention to the position of the block it is trying to pick up (the *source* block), as well as the position of the block it is trying to place on top of (the *target* block). Therefore, a properly trained network can learn to match the current state with the corresponding stage in the demonstration, and infer the identities of the source and target blocks expressed as soft attention weights over different blocks, which are then used to extract the corresponding positions to be passed to the manipulation network. Although we do not enforce this interpretation in training, our experiment analysis supports this interpretation of how the learned policy works internally.

## 4.3  Manipulation network

The manipulation network is the simplest component. After extracting the information of the source and target blocks, it computes the action needed to complete the current stage of stacking one block on top of another one, using a simple MLP network.[1] This division of labor opens up the possibility of modular training: the manipulation network may be trained to complete this simple procedure, without knowing about demonstrations or more than two blocks present in the environment. We leave this possibility for future work.

# 5  Experiments

We conduct experiments with the block stacking tasks described in Section 3.2.[2] These experiments are designed to answer the following questions:

- How does training with behavioral cloning compare with DAGGER?
- How does conditioning on the entire demonstration compare to conditioning on the final state, even when it already has enough information to fully specify the task?
- How does conditioning on the entire demonstration compare to conditioning on a "snapshot" of the trajectory, which is a small subset of frames that are most informative?

- Can our framework generalize to tasks that it has never seen during training?

To answer these questions, we compare the performance of the following architectures:

- **BC**: We use the same architecture as previous, but and the policy using behavioral cloning.
- **DAGGER**: We use the architecture described in the previous section, and train the policy using DAGGER.
- **Final state**: This architecture conditions on the final state rather than on the entire demonstration trajectory. For the block stacking task family, the final state uniquely identifies the task, and there is no need for additional information. However, a full trajectory, one which contains information about intermediate stages of the task's solution, can make it easier to train the optimal policy, because it could learn to rely on the demonstration directly, without needing to memorize the intermediate steps into its parameters. This is related to the way in which reward shaping can significantly affect performance in reinforcement learning [38]. A comparison between the two conditioning strategies will tell us whether this hypothesis is valid. We train this policy using DAGGER.
- **Snapshot**: This architecture conditions on a "snapshot" of the trajectory, which includes the last frame of each stage along the demonstration trajectory. This assumes that a segmentation of the demonstration into multiple stages is available at test time, which gives it an unfair advantage compared to the other conditioning strategies. Hence, it may perform better than conditioning on the full trajectory, and serves as a reference, to inform us whether the policy conditioned on the entire trajectory can perform as well as if the demonstration is clearly segmented. Again, we train this policy using DAGGER.

We evaluate the policy on tasks seen during training, as well as tasks unseen during training. Note that generalization is evaluated at multiple levels: the learned policy not only needs to generalize to new configurations and new demonstrations of tasks seen already, but also needs to generalize to new tasks.

Concretely, we collect 140 training tasks, and 43 test tasks, each with a different desired layout of the blocks. The number of blocks in each task can vary between 2 and 10. We collect 1000 trajectories per task for training, and maintain a separate set of trajectories and initial configurations to be used for evaluation. The trajectories are collected using a hard-coded policy.

## 5.1 Performance Evaluation

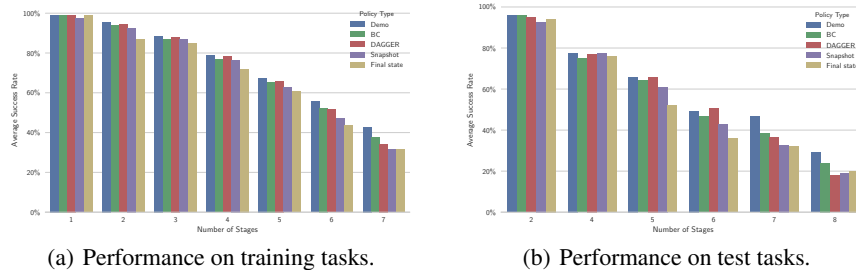

(a) Performance on training tasks.      (b) Performance on test tasks.

Figure 3: Comparison of different conditioning strategies. The darkest bar shows the performance of the hard-coded policy, which unsurprisingly performs the best most of the time. For architectures that use temporal dropout, we use an ensemble of 10 different downsampled demonstrations and average the action distributions. Then for all architectures we use the greedy action for evaluation.

Fig. 3 shows the performance of various architectures. Results for training and test tasks are presented separately, where we group tasks by the number of stages required to complete them. This is because tasks that require more stages to complete are typically more challenging. In fact, even our scripted policy frequently fails on the hardest tasks. We measure success rate per task by executing the greedy policy (taking the most confident action at every time step) in 100 different configurations, each conditioned on a different demonstration unseen during training. We report the average success rate over all tasks within the same group.

From the figure, we can observe that for the easier tasks with fewer stages, all of the different conditioning strategies perform equally well and almost perfectly. As the difficulty (number of stages) increases, however, conditioning on the entire demonstration starts to outperform conditioning on the final state. One possible explanation is that when conditioned only on the final state, the policy may struggle about which block it should stack first, a piece of information that is readily accessible from demonstration, which not only communicates the task, but also provides valuable information to help accomplish it.

More surprisingly, conditioning on the entire demonstration also seems to outperform conditioning on the snapshot, which we originally expected to perform the best. We suspect that this is due to the regularization effect introduced by temporal dropout, which effectively augments the set of demonstrations seen by the policy during training.

Another interesting finding was that training with behavioral cloning has the same level of performance as training with DAGGER, which suggests that the entire training procedure could work without requiring interactive supervision. In our preliminary experiments, we found that injecting noise into the trajectory collection process was important for behavioral cloning to work well, hence in all experiments reported here we use noise injection. In practice, such noise can come from natural human-induced noise through tele-operation, or by artificially injecting additional noise before applying it on the physical robot.

## 5.2 Visualization

We visualize the attention mechanisms underlying the main policy architecture to have a better understanding about how it operates. There are two kinds of attention we are mainly interested in, one where the policy attends to different time steps in the demonstration, and the other where the policy attends to different blocks in the current state. Fig. 4 shows some of the attention heatmaps.

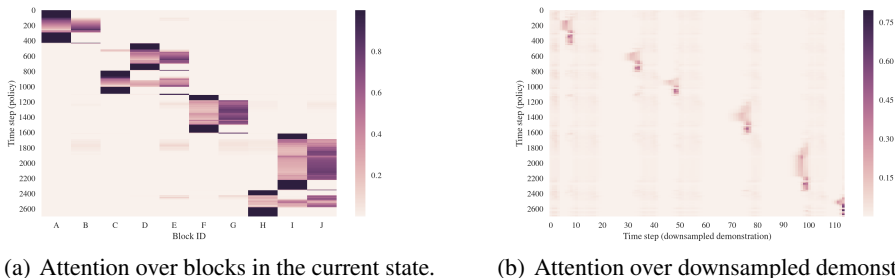

(a) Attention over blocks in the current state.      (b) Attention over downsampled demonstration.

Figure 4: Visualizing attentions performed by the policy during an entire execution. The task being performed is `ab cde fg hij`. Note that the policy has multiple query heads for each type of attention, and only one query head per type is visualized. (a) We can observe that the policy almost always focuses on a small subset of the block positions in the current state, which allows the manipulation network to generalize to operations over different blocks. (b) We can observe a sparse pattern of time steps that have high attention weights. This suggests that the policy has essentially learned to segment the demonstrations, and only attend to important key frames. Note that there are roughly 6 regions of high attention weights, which nicely corresponds to the 6 stages required to complete the task.

## 6 Conclusions

In this work, we presented a simple model that maps a single successful demonstration of a task to an effective policy that solves said task in a new situation. We demonstrated effectiveness of this approach on a family of block stacking tasks. There are a lot of exciting directions for future work. We plan to extend the framework to demonstrations in the form of image data, which will allow more end-to-end learning without requiring a separate perception module. We are also interested in enabling the policy to condition on multiple demonstrations, in case where one demonstration does not fully resolve ambiguity in the objective. Furthermore and most importantly, we hope to scale up

our method on a much larger and broader distribution of tasks, and explore its potential towards a general robotics imitation learning system that would be able to achieve an overwhelming variety of tasks.

# 7 Acknowledgement

We would like to thank our colleagues at UC Berkeley and OpenAI for insightful discussions. This research was funded in part by ONR through a PECASE award. Yan Duan was also supported by a Huawei Fellowship. Jonathan Ho was also supported by an NSF Fellowship.

## Footnotes

[1]In principle, one can replace this module with an RNN module. But we did not find this necessary for the tasks we consider.

[2]Additional experiment results are available in the Appendix, including a simple illustrative example of particle reaching tasks and further analysis of block stacking

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
