[Supplementary Material]

# 1 Illustrative Example: Particle Reaching

The particle reaching problem is a very simple family of tasks. In each task, we control a point robot to reach a specific landmark, and different tasks are identified by different landmarks. As illustrated in Fig. 1, one task could be to reach the orange square, and another task could be to reach the green triangle. The agent receives its own 2D location, as well as the 2D locations of each of the landmarks. Within each task, the initial position of the agent, as well as the positions of all the landmarks, can vary across different instances of the task.

Without a demonstration, the robot does not know which landmark it should reach, and will not be able to accomplish the task. Hence, this setting already gets at the essence of one-shot imitation, namely to communicate the task via a demonstration. After learning, the agent should be able to identify the target landmark from the demonstration, and reach the same landmark in a new instance of the task.

Figure 1: The robot is a point mass controlled with 2-dimensional force. The family of tasks is to reach a target landmark. The identity of the landmark differs from task to task, and the model has to figure out which target to pursue based on the demonstration. (left) illustration of the robot; (middle) the task is to reach the orange box, (right) the task is to reach the green triangle.

We consider three architectures for this problem:

- **Plain LSTM:** The first architecture is a simple LSTM with $512$ hidden units. It reads the demonstration trajectory, the output of which is then concatenated with the current state, and fed to a multi-layer perceptron (MLP) to produce the action.

- **LSTM with attention:** In this architecture, the LSTM outputs a weighting over the different landmarks from the demonstration sequence. Then, it applies this weighting in the test scene, and produces a weighted combination over landmark positions given the current state. This 2D output is then concatenated with the current agent position, and fed to an MLP to produce the action.

- **Final state with attention:** Rather than looking at the entire demonstration trajectory, this architecture only looks at the final state in the demonstration (which is already sufficient to communicate the task), and produce a weighting over landmarks. It then proceeds like the previous architecture.

Notice that these three architectures are increasingly more specialized to the specific particle reaching setting, which suggests a potential trade-off between expressiveness and generalizability.

The experiment results are shown in Fig. 2. We observe that as the architecture becomes more specialized, we achieve much better generalization performance. For this simple task, it appears that conditioning on the entire demonstration hurts generalization performance, and conditioning on just the final state performs the best even without explicit regularization. This makes intuitive sense, since the final state already sufficiently characterizes the task at hand.

However, the same conclusion does not appear to hold as the task becomes more complicated, as shown by the block stacking tasks in the main text.

Fig. 3 shows the learning curves for the three architectures designed for the particle reaching tasks, as the number of landmarks is varied, by running the policies over $100$ different configurations, and computing success rates over both training and test data. We can clearly observe that both LSTM-based architectures exhibit overfitting as the number of landmarks increases. On the other hand, using attention clearly improves generalization performance, and when conditioning on only the final state, it achieves perfect generalization in all scenarios. It is also interesting to observe that

Figure 2: Success rates of different architectures for particle reaching. The "Train" curves show the success rates when conditioned on demonstrations seen during training, and running the policy on initial conditions seen during training, while the "Test" curves show the success rates when conditioned on new trajectories and operating in new situations. Both attention-based architectures achieve perfect training success rates, and the curves are overlapped.

learning undergoes a phase transition. Intuitively, this may be when the network is learning to infer the task from the demonstration. Once this is finished, the learning of control policy is almost trivial.

Table 1 and Table 2 show the exact performance numbers for reference.

| #Landmarks | Plain LSTM | LSTM with attention | Final state with attention |
|:---:|:---:|:---:|:---:|
| 2 | 100.0% | 100.0% | 100.0% |
| 3 | 100.0% | 100.0% | 100.0% |
| 4 | 100.0% | 100.0% | 100.0% |
| 5 | 100.0% | 100.0% | 100.0% |
| 6 | 99.0% | 100.0% | 100.0% |
| 7 | 100.0% | 100.0% | 100.0% |
| 8 | 100.0% | 100.0% | 100.0% |
| 9 | 100.0% | 100.0% | 100.0% |
| 10 | 91.9% | 100.0% | 100.0% |

Table 1: Success rates of particle reaching conditioned on seen demonstrations, and running on seen initial configurations.

## 2 Further Details on Block Stacking

### 2.1 Full Description of Architecture

We now specify the architecture in pseudocode. We omit implementation details which involve handling a minibatch of demonstrations and observation-action pairs, as well as necessary padding and masking to handle data of different dimensions. We use weight normalization with data-dependent initialization Salimans and Kingma [2016] for all dense and convolution operations.

(a) Plain LSTM (Train)

(b) Plain LSTM (Test)

(c) LSTM with attention (Train)

(d) LSTM with attention (Test)

(e) Final state with attention (Train)

(f) Final state with attention (Test)

Figure 3: Learning curves for particle reaching tasks. Shown success rates are moving averages of past 10 epochs for smoother curves. Each policy is trained for up to 1000 epochs, which takes up to an hour using a Titan X Pascal GPU (as can be seen from the plot, most experiments can be finished sooner).

### 2.1.1 Demonstration Network

Assume that the demonstration has $T$ time steps and we have $B$ blocks. Our architecture only make use of the observations in the input demonstration but not the actions. Each observation is a $(3B + 2)$-dimensional vector, containing the $(x, y, z)$ coordinates of each block relative to the current position of the gripper, as well as a 2-dimensional gripper state indicating whether it is open or closed.

The full sequence of operations is given in Module 1. We first apply temporal dropout as described in the main text. Then we split the observation into information about the block and information about the robot, where the first dimension is time and the second dimension is the block ID. The robot state is broadcasted across different blocks. Hence the shape of outputs should be $\tilde{T} \times B \times 3$ and $\tilde{T} \times B \times 2$, respectively.

Then, we perform a $1 \times 1$ convolution over the block states to project them to the same dimension as the per-block embedding. Then we perform a sequence of neighborhood attention operations and $1 \times 1$ convolutions, where the input to the convolution is the concatenation of the attention result,

**Module 1** Demonstration Network

---

**Input:** Demonstration $d \in \mathbb{R}^{T \times (3B+2)}$
**Hyperparameters:** $p = 0.95$, $D = 64$
**Output:** Demonstration embedding $\in \mathbb{R}^{\tilde{T} \times B \times D}$, where $\tilde{T} = \lceil T(1-p) \rceil$ is the length of the downsampled trajectory.

```
d' ← TemporalDropout(d, probability=p)
block_state, robot_state ← Split(d')
h ← Conv1D(block_state, kernel_size=1, channels=D)
```
**for** $a \in \{1, 2, 4, 8\}$ **do**
    // Residual connections
```
    h' ← ReLU(h)
    attn_result ← NeighborhoodAttention(h')
    h' ← Concat({h', block_state, robot_state}, axis=-1)
    h' ← Conv1D(h', kernel_size=2, channels=D, dilation=a)
    h' ← ReLU(h')
    h ← h + h'
```
**end for**
```
demo_embedding ← h
```

---

the current block position, and the robot's state. This allows each block to query the state of other blocks, and reason about the query result in comparison with its own state and the robot's state. We use residual connections during this procedure.

### 2.1.2 Context Network

The pseudocode is shown in Module 2. We perform a series of attention operations over the demonstration, followed by attention over the current state, and we apply them repeatedly through an LSTM with different weights per time step (we found this to be slightly easier to optimize). Then, in the end we apply a final attention operation which produces a fixed-dimensional embedding independent of the length of the demonstration or the number of blocks in the environment.

### 2.1.3 Manipulation Network

Given the context embedding, this module is simply a multilayer perceptron. Pseudocode is given in Module 3.

## 2.2 Evaluating Permutation Invariance

During training and in the previous evaluations, we only select one task per equivalence class, where two tasks are considered equivalent if they are the same up to permuting different blocks. This is based on the assumption that our architecture is invariant to permutations among different blocks. For example, if the policy is only trained on the task abcd, it should perform well on task dcba, given a single demonstration of the task dcba. We now experimentally verify this property by fixing a training task, and evaluating the policy's performance under all equivalent permutations of it. As Fig. 4 shows, although the policy has only seen the task abcd, it achieves the same level of performance on all other equivalent tasks.

## 2.3 Effect of Ensembling

We now evaluate the importance of sampling multiple downsampled demonstrations during evaluation. Fig. 5 shows the performance across all training and test tasks, as the number of ensembles varies from 1 to 20. We observe that more ensembles helps the most for tasks with fewer stages. On the other hand, it consistently improves performance for the harder tasks, although the gap is smaller. We suspect that this is because the policy has learned to attend to frames in the demonstration trajectory where the blocks are already stacked together. In tasks with only 1 stage, for example, it is very easy for these frames to be dropped in a single downsampled demonstration. On the other hand, in tasks with more stages, it becomes more resilient to missing frames. Using more than 10

---

**Module 2** Context Network

---

**Input:** Demonstration embedding $h_{in} \in \mathbb{R}^{\tilde{T} \times B \times D}$, current state $s \in \mathbb{R}^{3B+2}$
**Hyperparameters:** $D = 64, t_{lstm} = 4, H = 2$
**Output:** Context embedding $\in \mathbb{R}^{2+6H}$

  // Split the current state into block state $\in \mathbb{R}^{B \times 3}$ and robot state broadcasted to all blocks $\in \mathbb{R}^{B \times 2}$
  `block_state, robot_state ← SplitSingle(s)`
  // Initialize LSTM output $\in \mathbb{R}^{B \times D}$ and state (including hidden and cell state) $\in \mathbb{R}^{B \times 2D}$
  `output, state ← InitLSTMState(size=B, hidden_dim=D)`
  **for** $\mathtt{t} = 1$ to $t_{lstm}$ **do**
    // Temporal attention: every block attend to the same time step
    `x ← output`
    **if** $\mathtt{t} > 1$ **then**
      `x ← ReLU(x)`
    **end if**
    // Computing query for attention over demonstration $\in \mathbb{R}^{B \times D}$
    `q ← Dense(x, output_dim=D)`
    // Compute result from attention $\in \mathbb{R}^{H \times B \times D}$
    `temp ← SoftAttention(query=q, context=h_in, memory=h_in, num_heads=H)`
    // Reorganize result into shape $B \times (HD)$
    `temp ← Reshape(Transpose(temp, (1, 0, 2)), (B, H*D))`

    // Spatial attention: each block attend to a different block separately
    `x ← output`
    **if** $\mathtt{t} > 1$ **then**
      `x ← ReLU(x)`
    **end if**
    `x ← Concat({x, temp}, axis=-1)`
    // Computing context for attention over current state $\in \mathbb{R}^{B \times D}$
    `ctx ← Dense(x, output_dim=D)`
    // Computing query for attention over current state $\in \mathbb{R}^{B \times D}$
    `q ← Dense(x, output_dim=D)`
    // Computing memory for attention over current state $\in \mathbb{R}^{B \times (HD+3)}$
    `mem ← Concat({block_state, temp}, axis=-1)`
    // Compute result from attention $\in \mathbb{R}^{B \times H \times (HD+3)}$
    `spatial ← SoftAttention(query=q, context=ctx, memory=mem, num_heads=H)`
    // Reorganize result into shape $B \times H(HD + 3)$
    `spatial ← Reshape(spatial, (B, H*(H*D+3)))`
    // Form input to the LSTM cell $\in \mathbb{R}^{B \times (H(HD+3)+HD+8)}$
    `input ← Concat({robot_state, block_state, spatial, temp}, axis=-1)`
    // Run one step of an LSTM with untied weights (meaning that we use different weights per time step
    `output, state ← LSTMOneStep(input=input, state=state)`
  **end for**
  // Final attention over the current state, compressing an $O(B)$ representation down to $O(1)$
  // Compute the query vector. We use a fixed, trainable query vector independent of the input data, with size $\in \mathbb{R}^{2 \times D}$ (we use two queries, originally intended to have one for the source block and one for the target block)
  `q ← GetFixedQuery()`
  // Get attention result, which should be of shape $2 \times H \times 3$
  `r ← SoftAttention(query=q, context=output, memory=block_state, num_heads=H)`
  // Form the final context embedding (we pick the first robot state since no need to broadcast here)
  `context_embedding ← Concat({robot_state[0], Reshape(r, 2*H*3)})`

---

---
**Module 3** Manipulation Network
---
**Input:** Context embedding $h_{in} \in \mathbb{R}^{2+6H}$
**Hyperparameters:** $H = 2$
**Output:** Predicted action distribution $\in \mathbb{R}^{|A|}$
```
h ← ReLU(Dense(h_in, output_dim=256))
h ← ReLU(Dense(h, output_dim=256))
action_dist ← Dense(h, output_dim=|A|)
```
---

Figure 4: Performance of policy on a set of tasks equivalent up to permutations.

ensembles appears to provide no significant improvements, and hence we used 10 ensembles in our main evaluation.

Figure 5: Performance of various number of ensembles.

## 2.4 Breakdown of Failure Cases

To understand the limitations of the current approach, we perform a breakdown analysis of the failure cases. We consider three failure scenarios: "Wrong move" means that the policy has arranged a layout incompatible with the desired layout. This could be because the policy has misinterpreted the demonstration, or due to an accidental bad move that happens to scramble the blocks into the wrong layout. "Manipulation failure" means that the policy has made an irrecoverable failure, for example if the block is shaken off the table, which the current hard-coded policy does not know how to handle. "Recoverable failure" means that the policy runs out of time before finishing the task, which may be due to an accidental failure during the operation that would have been recoverable given more time. As shown in Fig. 6, conditioning on only the final state makes more wrong moves compared to other architectures. Apart from that, most of the failure cases are actually due to manipulation failures that are mostly irrecoverable.[1] This suggests that better manipulation skills need to be acquired to make the learned one-shot policy more reliable.

Figure 6: Breakdown of the success and failure scenarios. The area that each color occupies represent the ratio of the corresponding scenario.

## 2.5 Learning Curves

Fig. 7 shows the learning curves for different architectures designed for the block stacking tasks. These learning curves do not reflect final performance: for each evaluation point, we sample tasks and demonstrations from training data, reset the environment to the starting point of some particular stage (so that some blocks are already stacked), and only run the policy for up to one stage. If the training algorithm is DAGGER, these sampled trajectories are annotated and added to the training set. Hence this evaluation does not evaluate generalization. We did not perform full evaluation as training proceeds, because it is very time consuming: each evaluation requires tens of thousands of

trajectories across over $> 100$ tasks. However, these figures are still useful to reflect some relative trend.

From these figures, we can observe that while conditioning on full trajectories gives the best performance which was shown in the main text, it requires much longer training time, simply because conditioning on the entire demonstration requires more computation. In addition, this may also be due to the high variance of the training process due to downsampling demonstrations, as well as the fact that the network needs to learn to properly segment the demonstration. It is also interesting that conditioning on snapshots seems to learn faster than conditioning on just the final state, which again suggests that conditioning on intermediate information is helpful, not only for the final policy, but also to facilitate training. We also observe that learning happens most rapidly for the initial stages, and much slower for the later stages, since manipulation becomes more challenging in the later stages. In addition, there are fewer tasks with more stages, and hence the later stages are not sampled as frequently as the earlier stages during evaluation.

## 2.6 Exact Performance Numbers

Exact performance numbers are presented for reference:

- Table 3 and Table 4 show the success rates of different architectures on training and test tasks, respectively;
- Table 5 shows the success rates across all tasks as the number of ensembles is varied;
- Table 6 shows the success rates of tasks that are equivalent to abcd up to permutations;
- Table 7, Table 8, Table 9, Table 10, and Table 11 show the breakdown of different success and failure scenarios for all considered architectures.

## 2.7 More Visualizations

Fig. 8 and Fig. 9 show the full set of heatmaps of attention weights. Interestingly, in Fig. 8, we observe that rather than attending to two blocks at a time, as we originally expected, the policy has learned to mostly attend to only one block at a time. This makes sense because during each of the grasping and the placing phase of a single stacking operation, the policy needs to only pay attention to the single block that the gripper should aim towards. For context, Fig. 10 and Fig. 11 show key frames of the neural network policy executing the task.

## Footnotes

[1]Note that the actual ratio of misinterpreted demonstrations may be different, since the runs that have caused a manipulation failure could later lead to a wrong move, were it successfully executed. On the other hand, by visually inspecting the videos, we observed that most of the trajectories categorized as "Wrong Move" are actually due to manipulation failures (except for policy conditioning on the final state, which does seem to occasionally execute an actual wrong move).

## References

Tim Salimans and Diederik P Kingma. Weight normalization: A simple reparameterization to accelerate training of deep neural networks. In *Advances in Neural Information Processing Systems*, pages 901–901, 2016.

| #Landmarks | Plain LSTM | LSTM with attention | Final state with attention |
|---|---|---|---|
| 2 | **100.0**% | **100.0**% | **100.0**% |
| 3 | **100.0**% | **100.0**% | **100.0**% |
| 4 | 99.0% | **100.0**% | **100.0**% |
| 5 | 98.0% | **100.0**% | **100.0**% |
| 6 | 99.0% | **100.0**% | **100.0**% |
| 7 | 98.0% | **100.0**% | **100.0**% |
| 8 | 93.9% | 99.0% | **100.0**% |
| 9 | 83.8% | 94.9% | **100.0**% |
| 10 | 50.5% | 85.9% | **100.0**% |

Table 2: Success rates of particle reaching conditioned on unseen demonstrations, and running on unseen initial configurations.

| #Stages | Demo | DAGGER | BC | Snapshot | Final state |
|---|---|---|---|---|---|
| 1 | **99.1**% | **99.1**% | **99.1**% | 97.2% | 98.8% |
| 2 | **95.6**% | **94.3**% | 93.7% | 92.6% | 86.7% |
| 3 | **88.5**% | **88.0**% | 86.9% | 86.7% | 84.8% |
| 4 | **78.6**% | **78.2**% | 76.7% | 76.4% | 71.9% |
| 5 | **67.3**% | **65.9**% | 65.4% | 62.5% | 60.6% |
| 6 | **55.7**% | 51.5% | **52.4**% | 47.0% | 43.6% |
| 7 | **42.8**% | 34.3% | **37.5**% | 31.4% | 31.5% |

Table 3: Success rates of different architectures on training tasks of block stacking.

| #Stages | Demo | DAGGER | BC | Snapshot | Final state |
|---|---|---|---|---|---|
| 2 | 95.8% | 94.9% | **95.9**% | 92.8% | 94.1% |
| 4 | **77.6**% | 77.0% | 74.8% | **77.2**% | 75.8% |
| 5 | **65.9**% | **65.9**% | 64.3% | 61.1% | 51.9% |
| 6 | 49.4% | **50.6**% | 46.5% | 42.6% | 35.9% |
| 7 | **46.5**% | 36.5% | **38.5**% | 32.8% | 32.0% |
| 8 | **29.0**% | 18.0% | **24.0**% | 19.0% | 20.0% |

Table 4: Success rates of different architectures on test tasks of block stacking.

| #Stages | 1 Ens. | 2 Ens. | 5 Ens. | 10 Ens. | 20 Ens. |
|---|---|---|---|---|---|
| 1 | 91.9% | 95.4% | 98.8% | **99.1**% | 98.7% |
| 2 | 92.3% | 92.2% | 94.5% | **94.6**% | 94.1% |
| 3 | 86.0% | 86.8% | 87.9% | **88.0**% | 87.9% |
| 4 | 76.6% | 77.4% | 77.9% | 78.0% | **78.3**% |
| 5 | 65.1% | 65.0% | 65.3% | **65.9**% | 65.5% |
| 6 | 49.0% | 50.4% | 50.1% | **51.3**% | 50.8% |
| 7 | 34.4% | 36.1% | 36.0% | 34.9% | **36.8**% |
| 8 | 20.0% | **21.0**% | **21.0**% | 18.0% | 20.0% |

Table 5: Success rates of varying number of ensembles using the DAGGER policy conditioned on full trajectories, across both training and test tasks.

| Task ID | Success Rate |
|---------|--------------|
| abcd | 83.0% |
| abdc | 86.0% |
| acbd | 92.0% |
| acdb | 84.0% |
| adbc | 91.0% |
| adcb | 88.0% |
| bacd | 92.0% |
| badc | 90.0% |
| bcad | 92.0% |
| bcda | 88.0% |
| bdac | 94.0% |
| bdca | 88.0% |
| cabd | 82.0% |
| cadb | 87.0% |
| cbad | 95.0% |
| cbda | 87.0% |
| cdab | 91.0% |
| cdba | 93.0% |
| dabc | 90.0% |
| dacb | 92.0% |
| dbac | 88.0% |
| dbca | 90.0% |
| dcab | 91.0% |
| dcba | 84.0% |

Table 6: Success rates of a set of tasks that are equivalent up to permutations, using the DAGGER policy conditioned on full trajectories.

| #Stages | Success | Recoverable failure | Manipulation failure | Wrong move |
|---------|---------|---------------------|----------------------|------------|
| 1 | 99.3% | 0.0% | 0.7% | 0.0% |
| 2 | 95.9% | 0.4% | 3.7% | 0.0% |
| 3 | 89.1% | 0.7% | 10.1% | 0.1% |
| 4 | 79.2% | 1.2% | 19.4% | 0.1% |
| 5 | 67.5% | 1.4% | 30.9% | 0.2% |
| 6 | 55.2% | 1.4% | 43.1% | 0.3% |
| 7 | 44.6% | 1.7% | 53.2% | 0.6% |
| 8 | 30.9% | 4.3% | 64.9% | 0.0% |

Table 7: Breakdown of success and failure scenarios for Demo policy.

| #Stages | Success | Recoverable failure | Manipulation failure | Wrong move |
|---------|---------|---------------------|----------------------|------------|
| 1 | 99.4% | 0.0% | 0.6% | 0.0% |
| 2 | 95.3% | 0.9% | 3.8% | 0.0% |
| 3 | 89.1% | 1.9% | 8.8% | 0.1% |
| 4 | 79.5% | 3.5% | 16.7% | 0.3% |
| 5 | 69.1% | 5.0% | 25.6% | 0.3% |
| 6 | 55.8% | 7.3% | 36.4% | 0.5% |
| 7 | 39.0% | 8.6% | 51.5% | 0.8% |
| 8 | 21.2% | 14.1% | 62.4% | 2.4% |

Table 8: Breakdown of success and failure scenarios for DAGGER policy.

(a) All Stages

(b) Stage 0

(c) Stage 1

(d) Stage 2

(e) Stage 3

(f) Stage 4

(g) Stage 5

(h) Stage 6

Figure 7: Learning curves of block stacking task. The first plot shows the average success rates over initial configurations of all stages. The subsequent figures shows the breakdown of each stage. For instance, "Stage 3" means that the first 3 stacking operations are already completed, and the policy is evaluated on its ability to perform the 4th stacking operation.

| #Stages | Success | Recoverable failure | Manipulation failure | Wrong move |
|---------|---------|---------------------|----------------------|------------|
| 1 | 99.6% | 0.0% | 0.4% | 0.0% |
| 2 | 95.6% | 1.1% | 3.2% | 0.1% |
| 3 | 88.1% | 2.2% | 9.5% | 0.2% |
| 4 | 78.5% | 4.5% | 16.8% | 0.2% |
| 5 | 67.2% | 6.6% | 25.7% | 0.4% |
| 6 | 53.9% | 8.3% | 37.1% | 0.6% |
| 7 | 40.6% | 9.8% | 48.7% | 0.9% |
| 8 | 27.0% | 13.5% | 58.4% | 1.1% |

Table 9: Breakdown of success and failure scenarios for BC policy.

| #Stages | Success | Recoverable failure | Manipulation failure | Wrong move |
|---------|---------|---------------------|----------------------|------------|
| 1 | 99.1% | 0.0% | 0.9% | 0.0% |
| 2 | 94.5% | 1.6% | 3.8% | 0.1% |
| 3 | 88.0% | 2.5% | 9.3% | 0.2% |
| 4 | 78.9% | 4.6% | 16.2% | 0.3% |
| 5 | 65.6% | 8.0% | 25.8% | 0.6% |
| 6 | 50.8% | 8.3% | 40.2% | 0.7% |
| 7 | 36.1% | 9.2% | 54.2% | 0.4% |
| 8 | 21.6% | 11.4% | 65.9% | 1.1% |

Table 10: Breakdown of success and failure scenarios for Snapshot policy.

| #Stages | Success | Recoverable failure | Manipulation failure | Wrong move |
|---------|---------|---------------------|----------------------|------------|
| 1 | 99.2% | 0.0% | 0.8% | 0.0% |
| 2 | 95.1% | 1.3% | 3.6% | 0.0% |
| 3 | 86.7% | 2.5% | 9.7% | 1.1% |
| 4 | 75.2% | 4.0% | 18.3% | 2.5% |
| 5 | 60.5% | 4.3% | 31.2% | 4.0% |
| 6 | 45.5% | 4.7% | 45.5% | 4.3% |
| 7 | 34.9% | 5.6% | 57.3% | 2.2% |
| 8 | 24.1% | 3.6% | 72.3% | 0.0% |

Table 11: Breakdown of success and failure scenarios for Final state policy.

(a) Head 0

(b) Head 1

(c) Head 2

(d) Head 3

Figure 8: Heatmap of attention weights over different blocks of all 4 query heads.

(a) Head 0

(b) Head 1

(c) Head 2

(d) Head 3

(e) Head 4

(f) Head 5

Figure 9: Heatmap of attention weights over downsampled demonstration trajectory of all 6 query heads. There are 2 query heads per step of LSTM, and 3 steps of LSTM are performed.

Figure 10: Illustration of the task used for the visualization of attention heatmaps (first half). The task is `ab cde fg hij`. The left side shows the key frames in the demonstration. The right side shows how, after seeing the entire demonstration, tthe policy reproduces the same layout in a new initialization of the same task.

Figure 11: Illustration of the task used for the visualization of attention heatmaps (second half). The task is `ab cde fg hij`. The left side shows the key frames in the demonstration. The right side shows how, after seeing the entire demonstration, tthe policy reproduces the same layout in a new initialization of the same task.