[Reviews · NeurIPS 2017]

Reviewer 1



Summary --- Complex and useful robotic manipulation tasks are difficult because of the difficulty of manipulation itself, but also because it's difficult to communicate the intent of a task. Both of these problems can be alleviated through the use of imitation learning, but in order for this to be practical the learner must be able to generalize from few examples. This paper presents an architecture inspired by recent work in meta learning which generalizes manipulation of a robot arm from a single task demonstration; i.e., it does one-shot imitation learning. The network is something like a seq2seq model that uses multiple attention mechanisms in the style of "Neural Machine Translation by Jointly Learning to Align and Translate". There is a demonstration network, a context network and a manipulation network. The first two produce a context embedding, which is fed to a simple MLP (the manipulation network) that produces an action that tell the arm how to move the blocks. This context embedding encodes the demonstration sequence of states and the current state of arm in a way that selects only relevant blocks at relevant times from the demonstration. More specifically, the demonstration network takes in a sequence (hundreds of time steps long) of block positions and gripper states (open or closed) and embeds the sequence through a stack of residual 1d convolution block. At each layer in the stack and at each time step, an attention mechanism allows that layer to compare its blocks to a subset of relevant blocks. This sequence embedding is fed into the context network which uses two more attention mechanisms to first select the relevant period of the demonstration to the current state and then attend to relevant blocks of the demonstration. None of the aforementioned interpretations are enforced via supervision, but they are supported by manual inspection of attention distributions. The network is trained to build block towers that mimic those built in a demonstration in the final arrangement of blocks where the demonstration and test instance start from initially different configuration. DAGGER and Behavior Cloning (BC) are used to train the above architecture and performance is evaluated by measuring whether the correct configuration of blocks is produced or not. DAGGER and BC perform about the same. Other findings: * The largest difference in performance is due to increased difficulty. The more blocks that need to be stacked, the worse the desired state is replicated. * The method generalizes well to new tasks (number and height of towers), not just instances (labelings and positions of blocks). * The final demonstration state is enough to specify a task, so a strong baseline encodes only this state but performs significantly worse than the proposed model. Strengths --- This is a very nice paper. It proposes a new model for a well motivated problem based on a number of recent directions in neural net design. In particular, this proposes a novel combination of elements from few-shot learning, (esp meta learning approaches like Matching Networks), imitation learning, transfer learning, and attention models. Furthermore, some attention parts of the model (at least for examples provided) seem cleanly interpretable, as shown in figure 4. Comments/Suggestions/Weaknesses --- There aren't any major weaknesses, but there are some additional questions that could be answered and the presentation might be improved a bit. * More details about the hard-coded demonstration policy should be included. Were different versions of the hard-coded policy tried? How human-like is the hard-coded policy (e.g., how a human would demonstrate for Baxter)? Does the model generalize from any working policy? What about a policy which spends most of its time doing irrelevant or intentionally misleading manipulations? Can a demonstration task be input in a higher level language like the one used throughout the paper (e.g., at line 129)? * How does this setting relate to question answering or visual question answering? * How does the model perform on the same train data it's seen already? How much does it overfit? * How hard is it to find intuitive attention examples as in figure 4? * The model is somewhat complicated and its presentation in section 4 requires careful reading, perhaps with reference to the supplement. If possible, try to improve this presentation. Replacing some of the natural language description with notation and adding breakout diagrams showing the attention mechanisms might help. * The related works section would be better understood knowing how the model works, so it should be presented later.

Reviewer 2



The main contribution of the paper is a new architecture for the block stacking task.Although the results seem to look good, I think that the contribution in terms of insight and novelty that I gather from reading the paper ultimately seem insufficient. Since neural netaork architecture seems to be the factor that leads to good performance, I think it'd be nice to see how the same architecture would fare in some other domain as well?